# An Interpretable Answer Scoring Framework

Omar Alonso
Amazon
omralon@amazon.com

Preetam Prabhu Srikar Dammu*
University of Washington
preetams@uw.edu

Diji Yang*
University of California Santa Cruz
dyang39@ucsc.edu

## ABSTRACT

In this new LLM-world where users can ask any natural language question, the focus is on the generation of answers with reliable information while satisfying the original intent. LLMs are known to generate multiple versions of answers for the same question, some of which may be better than others. Identifying the most suitable response that adequately addresses the question is non-trivial. In order to tackle this problem, we propose an interpretable scoring system that considers three aspects of an answer: knowledge, content, and structure. We provide an answer quality score method that is explainable and can be a key signal to determining a good answer.

## 1 INTRODUCTION

The information-seeking process assumes an interaction cycle that includes the identification of an information need, a specification of such need in a query, the inspection of results, and if needed, a reformulation of the query [26]. Users may use a search engine, an intelligent assistant, or a ChatGPT-like system [17] to accomplish a specific task, but the cognitive model stays the same. The underlying system can provide better services, but from the user's perspective, the overall information access procedure remains constant.

Part of the search process involves assessing how relevant the results are according to the original intent [27]. There are many factors that contribute to relevance criteria and degrees of relevance. Instead of providing a link to a web page or document that contains relevant bits, LLMs-based systems provide an answer that synthesizes all the relevant information [1]. The verbosity of such models can, unfortunately, lead to statements that are incorrect and, therefore, produce wrong responses [10].

We propose an explicit and interpretable scoring system for the answer generated by the LLM, thereby providing a measure for the reliability of the LLM response. To derive such a score, we look at three main characteristics of an answer: *knowledge*, *structure*, and *content* and present the overall architecture of our proposed system in Figure 1. Specifically, the *knowledge component* evaluates the statements or information present in the answer against a knowledge graph (KG) for reliability. After identifying the entities present in a statement, all relevant entries are fetched from a suitable KG. We then evaluate if the corresponding statement

*Work done during internship at Amazon.

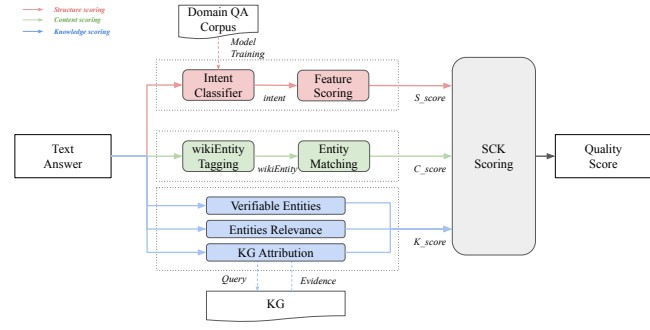

**Figure 1: Interpretable answer scoring architecture.**

is attributable or contradictory with respect to the retrieved entities by querying an LLM. Adjustable hyper-parameters are used to penalize contradictory statements and award correct statements. The *structure component* focuses on the gap between the ideal and generated answers. We first define a feature collection to match the features of the ideal answer for each question intent. We utilize embedding alignment techniques to capture the shared attributes between question-answer pairs, which are represented by the output distribution of the intent classifier. The final structure score is jointly determined by the feature collection and the learnable embedding projection. The *content component* is engineered to align important entity details from the question with those articulated in the answer. This alignment serves as a gauge for how effectively the answer mirrors the question. By identifying and comparing the key entities present in the question and the answer, we gauge the extent to which the answer addresses the essence of the inquiry.

The Interpretability is essential for building information retrieval and knowledge management systems [2]; on the other hand, in the era of large-scale language models (LLM), providing the interpretable score for the LLM generated answer is also crucial for the evaluation of LLMs' capabilities [7] as well as for providing meaningful feedback or reward for the model training [17, 24]. Therefore, we believe that defining reliable automatic scoring systems is a step in the right direction.

## 2 ANSWERING SCORING SYSTEM

Our work is based on identifying specific answer properties that provide signals and aggregate them into a final score. In general, we use knowledge graphs to compute the Knowledge scores (Sec. 2.1), question intent analyzing [4] to assist with the design of the Structure score (Sec. 2.2), and entity-based answer analyzing [15] to drive the Content score (Sec. 2.3). As shown in the Figure 1, the overall scoring pipeline considered the above three scores and functions in a fully automatic manner. This modularity is by design as it allows

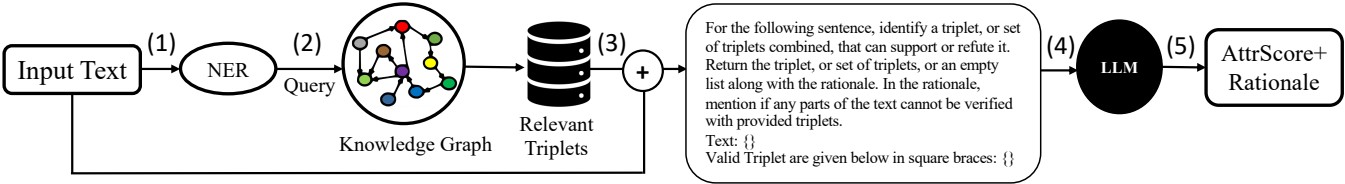

**Figure 2: Knowledge Graph Attribution Score (KG-AttrScore) Generation Pipeline.**

us to improve a specific component. Some of the proposed features are related to previous work on aboutness axioms [5, 6].

## 2.1 Knowledge Score

In addressing the knowledge limitations in LLMs, the use of information stored in KGs as an external source is beneficial. KGs are adaptable and suitable for handling evolving datasets, such as those in e-commerce. A KG organizes properties and relations as triplets and offers a clear way to organize information, facilitating efficient retrieval. The use of KGs reduces the need for direct one-to-one mapping of reference documents, enhancing the information processing task and potentially reducing errors associated with direct mapping methods.

*2.1.1 Verifiable Entities Score (VE-score).* Quantifies the ratio of entities that are verifiable by the KG.

$$VE\text{-}Score = |text\_entities \cap KG\_entities|/|text\_entities| \quad (1)$$

where $text\_entities$ = entities identified in the text and $KG\_entities$ = entities present in the KG. The $VE\text{-}Score$ measures the ratio of verifiable entities generated by the LLM. This penalizes the LLM when it hallucinates new entities that are invalid.

*2.1.2 Entities Relevance Score (ER-Score).* Quantifies the relevance of the verifiable entities.

$$ER\text{-}Score = \sum_{n=1}^{n} Jaccard(ques, entity\_desc_i) \\ + cosine\_similarity(e\_ques, e\_entity\_desc_i) \quad (2)$$

where each question has $n$ KG entities, and

- $entity\_desc_i$ = description of $i^{th}$ entity obtained from KG
- $e\_ques$ = text embedding vector of the question
- $e\_entity\_desc_i$= text embedding vector of the $i^{th}$ entity

*2.1.3 KG-based Attribution Score (KG-attrScore).* The proposed attribution score presents several advancements over previous attribution methods [19, 25]. Firstly, it offers a more granular approach by enabling sentence-level evaluation. This allows for a detailed analysis of the text, providing a finer understanding of how each sentence contributes to the overall context. Secondly, our approach is more comprehensive. Unlike previous methods that evaluate content against a single document, this approach utilizes a KG as the reference point. This broadens the scope of the evaluation, incorporating a wider array of information and relationships present within the KG, thus offering a more holistic assessment. Lastly, the attribution score is more dynamic due to its integration with the KG. As the KG is updated, the attribution score automatically adjusts to these changes. This ensures that the evaluation remains

current and reflects the latest information and trends, enhancing its relevance and accuracy over time.

Given a natural language query $q$, an answer $a$, and a set of triplets $SoT$, the scoring function $f$ takes (q, a, SoT) as input and outputs a decision along with the rationale. Specifically,

- f(q, a, SoT) = 1 (Attributable) + Rationale
- f(q, a, SoT) = 0 (Extrapolatory) + Rationale
- f(q, a, SoT) = -1 (Contradictory) + Rationale

In operationalizing the objective function, we employ an LLM Query. LLMs have shown proficiency in Natural Language Inference (NLI) tasks [8, 11, 18, 23], which forms the backbone of this approach.

The NLI task [9] involves assessing the relationship between a 'premise' and a 'hypothesis', which in our case corresponds to the set of triplets and Q&A pair. The goal is to determine if the hypothesis is true (attributable), false (contradiction), or cannot be determined (extrapolatory) based on the given premise. This process requires analyzing the semantic relationship between the premise and the hypothesis, a task for which LLMs are particularly well-suited due to their advanced capabilities in language understanding. The prompt for performing this operation is shown in Figure 2.

In our experiment, we utilized the Flan-T5 XXL, with a generation temperature set to 0.2, to ensure controlled and precise outputs. Although Flan-T5 XXL was employed in our implementation, the approach is flexible and can be adapted to use other models as needed. This adaptability allows the framework to be tailored to different domains and answer types.

```
SELECT ?subject ?predicate ?object
WHERE {{
wd:{subject_qid} ?predicate wd:{object_qid}.
BIND(wd:{subject_qid} AS ?subject).
BIND(wd:{object_qid} AS ?object).
}}
```
**SPARQL Query for retrieving triplets from Wikidata**

We utilize a SPARQL query to retrieve relevant triplets from the Wikidata KG [21]. For simplicity, our current approach focuses on concepts represented by direct links within the KG. This method can be further developed to accommodate more complex relationships. An example of the $KG - AttrScore$ for the LLM generated answer in Figure 3 is presented in Table 1.

*2.1.4 Final K-score.* We compute the final K-score that encapsulates the three sub-scores discussed above as follows

$$K-Score = \alpha * VE-score + \beta * ER-score + \gamma * KG-AttrScore \quad (3)$$

| Sent. # | Prediction | Relevant triplets | Rationale | Identified Wiki entities |
|---------|------------|-------------------|-----------|--------------------------|
| 1 | Supports | [('Barack Obama', 'position held', 'President of the United States')] | The triplet ('Barack Obama', 'position held', 'President of the United States') supports the statement that Barack Obama is a former President of the United States. | Q76, Q30, Q305178, Q1169 |
| 2 | Supports | [('Barack Obama', 'ethnic group', 'African Americans')] | The triplet ['Barack Obama', 'ethnic group', 'African Americans'] supports the statement that he is the first African American to have held the office. However, the triplets provided do not directly support the statement about his progressive policies and leadership. | Q484275, Q76, Q1156854, Q49085, Q12823105 |
| 3 | Supports | [('Patient Protection and Affordable Care Act', 'signatory', 'Barack Obama')] | The triplet ['Patient Protection and Affordable Care Act', 'signatory', 'Barack Obama'] supports the statement that Obama enacted landmark legislation such as the Affordable Care Act. However, it does not provide direct support for the other legislation mentioned or the issues addressed by Obama during his presidency. | Q43807777, Q191600, Q23789296, Q11690, Q1414593, Q76, Q1725430, Q17163140, Q7888423, Q7942, Q86447, Q700719 |
| 4 | Neutral | NA | NA | Q11471, Q118157, Q76, Q5152897, Q32518, Q11771944 |

**Table 1: Examples of KG Attribution Score (KG-AttrScore). KG is Wikidata. Sentence number (first column) refers to the respective sentence in Figure 3. Retrieved triplets are not displayed due to space restrictions.**

```
Q: Can you provide a brief introduction of Barack Obama?

A: "Sure, here's a brief introduction to Barack Obama: Barack
Obama is a former President of the United States. He is the
first African American to have held the office, and is widely
recognized for his progressive policies and leadership. During
his presidency, Obama enacted landmark legislation such as
the Affordable Care Act and the Dodd-Frank Wall Street Reform
and Consumer Protection Act, and worked to address issues
such as climate change, immigration reform, and civil rights.
Obama is widely respected for his intelligence, charisma, and
commitment to public service, and is considered one of the
most influential leaders of our time."
```

**Figure 3: LLM generated answer.**

The hyperparameters $\alpha$, $\beta$, and $\gamma$ can be fine-tuned to align with desirable answer characteristics and domain requirements. This flexibility allows the K-score to adapt to a variety of contexts and use cases. In our experiments, we assigned equal weightage to all three parameters ($\alpha = \beta = \gamma \approx 0.3$), which balanced the influence of each sub-score.

## 2.2 Structure Score

*Design.* Lehnert [14] highlights the importance of understanding question intent in question and answer systems to satisfy users' information needs. Even if both questions are about the information of the same entities, the different question intents result in two types of answer structures. Bolotova et al. [4] states that with a well-defined question intent taxonomy, the answers that satisfy one intent are expected to have a specific (language) structure. Thus, Bolotova et al. [4] proposes a comprehensive taxonomy, NFQA, and the expected structure of answers. Specifically, NFQA includes six categories for non-factoid questions: instruction, reason, evidence-based, comparison, experience, and debate. However, NFQA lacks the fine-grained feature of how each intent is different from others, which limits the

reliability and interpretability of the taxonomy. Moreover, it is hard to build an automatic scoring system based on the coarse-grained question description. Building on top of the NFQA framework, we have developed a specialized set of features tailored to our use case. As shown in our scoring pipeline, highlighted by a red line in Figure 1, encompasses both the fine-tuning of an intent classifier and the subsequent feature scoring. Given the domain-specific nature of the classifier, it undergoes fine-tuning within our target domain to ensure its efficacy. Upon successful training, the classifier's intent predictions could be used to guide the feature-scoring function. Table 2 shows the feature set to differentiate between various intents. We design a corresponding function for each feature to map feature representations to scores, thereby differentiating question intents and their respective answer structures. Practically, the feature set includes question length, which indicates complexity and the level of detail required for responses; tense analysis, which is closely connected to the question intent and helps in understanding the temporal context of inquiries; keyword extraction, enabling the identification of the primary subject matter of questions for efficient classification; pronoun detection, providing insights into the type of information sought, such as factual details or explanations; and consideration of the expected answer structure, such as lists or paragraphs, to match the format preferences of users, thereby enhancing precision in classification. By incorporating these features, we can effectively analyze and classify questions based on their underlying intents, facilitating more accurate and tailored responses to user inquiries.

*Implementation.* Following the standard rule-based syntactic parsing works [28], the above-mentioned functions are mainly supported by NLTK (The Natural Language Toolkit) [3]. Specifically, we first applied dependency parsing and POS (Part-of-Speech) tagging over the answer sentence compliance with the Penn Treebank [16]. Then, the length can be easily captured by counting the token, while the tense and pronoun can be decided by the predicate and its POS tag. The format of the answer is detected by the special tokens, e.g., a step-by-step list requires a newline token at the end

| Category | Length | Tense | Keyword | Pronoun | Format |
|---|---|---|---|---|---|
| Instruction | short to moderate | present; future | steps; follows | 2nd | step-by-step list |
| Reason | short to moderate | present; past | because; due to; since | 3rd, 2nd | sentences  short paragraph |
| Evidence-based | moderate to long | past | found; indicate | 3rd | paragraph |
| Comparison | moderate | present; past | different; compared to | 3rd | side-by-side list |
| Experience | moderate to long | past | noticed; felt | 1st | paragraphs |
| Debate | moderate to long | present; past | believe; think; | 1st, 3rd | point-counterpoint paragraphs |

**Table 2: Feature collection of NFQA taxonomy.**

of each step description. Finally, inspired by NFQA [4], we also manually define the keywords for each question intent and use exact match to provide the keyword overlap score.

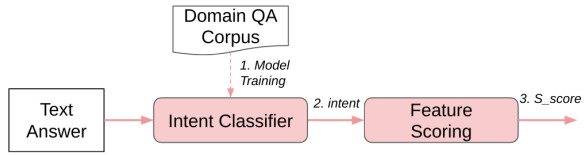

**Figure 4: Structure Scoring Pipeline**

## 2.3 Content Score

The Content Score, $C_{score}$, quantifies the relevance and completeness of an answer by measuring the overlap of entities that are important to the question's context. Specifically, for question-answering tasks that utilize Wikipedia data, these important entities are referred to as *wikiEntities*. The score is computed as the ratio of important entities found in the model-generated answer ($E$) to the total number of important entities in the ground-truth answer ($E_{total}$), represented mathematically as:

$$C_{score} = \frac{|E \cap E_{total}|}{|E_{total}|} \quad (4)$$

This scoring mechanism emphasizes the model's ability to identify and include all relevant entities in its response, ensuring that the answer is not only correct but also comprehensive. For instance, as depicted in Figure 5, when comparing three products—iPhone 13, Pixel 6, and Galaxy S23, captured by their respective wikiEntities Q108118280, Q108939091, and Q115127699—the Content Score is calculated based on the entity overlap. As the answer includes only two out of the three entities, the recall of important entities is 2/3, indicating that the response falls short of fully addressing the comparison request.

## 2.4 Final Quality Score

The overall scoring equation is shown as Equation 5, where the Quality score $SCK$ is the weighted sum of $S$, $C$, and $K$ scores.

$$SCK = \alpha * S\_score + \beta * C\_score + \gamma * K\_score \quad (5)$$

Since question intent faithfully reflects what kind of answer the user expects, the value of the weight factors are designed intentwise, i.e., the structure of the expected answers for different intent has varying feature weight. For example, Factoid intent is more

```
Q: Compare these iPhone 13, Pixel 6, and Galaxy S23

A: The iPhone 12, released in 2020, offers a sleek design,
a capable A14 Bionic chip, and a vibrant Super Retina
XDR display. The iPhone 13, introduced in 2021, builds
upon its predecessor with the more powerful A15 Bionic
chip, improved camera system, and longer battery life. The
Pixel 6, launched in 2021 as well, featuring impressive
computational photography capabilities, alongside a clean
Android experience. Ultimately, the choice between them
depends on user preferences, with iPhone models known for
their ecosystem integration and build quality, while the
Pixel 6 excels in camera innovation and a more pure Android
experience.
```

**Figure 5: LLM answer compares two of the three products. The underline highlights the Wikientities found in the text.**

| Intent | $\alpha$ | $\beta$ | $\gamma$ | Reason |
|---|---|---|---|---|
| Instruction | 0.5 | 0.2 | 0.3 | Answer structure > KG matching > entity mentioning |
| Reason | 0.5 | 0.3 | 0.2 | keyword from feature collection is useful |
| Evidence-based | 0.2 | 0.3 | 0.5 | No salient feature from structure, Knowledge more important |
| Comparison | 0.5 | 0.3 | 0.2 | Comparison structure is easy to capture |
| Experience | 0.4 | 0.5 | 0.1 | Entity matching > structure features > KG matching |
| Debate | 0.4 | 0.4 | 0.2 | Can hardly be grounded by KG |
| Factoid | 0.1 | 0.1 | 0.8 | KG matching > others |

**Table 3: Heuristic design for weight factors. The Reason column includes the rationale behind the design, and the Greater-than sign means that the former score is more important than the latter score under the current intent.**

concerned with matching knowledge (K score) than with the structure of the answer (S score); however, Experience intent should focus more on the common entity (C score) than on some piece of knowledge extracted from the KG. In this work, we present our heuristic-designed weight factors and the rationale behind them in Table 3. In the case that label data is available, the value of weight factors could also be learnable parameters.

| Question | Answer | Intent | S | C | K | SCK |
|---|---|---|---|---|---|---|
| What is the difference between soups and stews? | Stews are normally chunkier w/ larger more inexpensive cuts of meat since u simmer it on low heat for a long time. Most are made with a tomato base. while soups on the other hand don't require large cuts of meat and are perfect for a quick meal w/ little time and ingredients involved compared to stews. | Comparison | 1 | 1 | 0.42 | **0.88** |
| What's the application area for router? | The application area would be anywhere you want to connect two networks together. At home, connecting your local network to the internet network, definitely at businesses | Factoid | 0.25 | 0.2 | 0.74 | **0.63** |
| How many liquor stores in Houston? | Not enough if you can still formulate questions. :P | Experience | 0.0 | 0.0 | 0.10 | **0.01** |

**Table 4: Three representative examples from our scoring system. SCK is the final answer quality score.**

| Intent | Avg. SCK |
|---|---|
| Comparison | 0.39 |
| Debate | 0.31 |
| Evidence-based | 0.24 |
| Experience | 0.20 |
| Factoid | 0.29 |
| Instruction | 0.33 |
| Reason | 0.33 |

**Table 5: Average SCK score per intent for 1K data set.**

## 3 EXPERIMENTS AND RESULTS

The system is evaluated on 1K question-answer pairs, all of which were randomly selected from the YahooQA dataset[1], a corpus containing Non-factoid question-answering pairs. To the best of our knowledge, there is no available data set of LLM answers, so we use YahooQA as a human baseline to test our framework. To illustrate, we present three representative examples in Table 4. The structure of the first question's answer aligns well with the Comparison intent, resulting in a high structure score. Additionally, mentioning both key entities contributes to a high content score. Although not all information can be grounded, the weight factor defined for Comparison intent in Table 3 ensures that decent Structure (S) and Content (C) scores lead to an acceptable final quality score. In the second example, despite the absence of the key entity "router", the answer's overall quality score remains moderately high due to the emphasis on knowledge groundedness, which is crucial for Factoid questions. The last example with a 0.1 quality score indicates a poor response to the question, characterized by irrelevant content and a lack of substantial information. The average SCK scores distribution for the 1K per intent category is presented in Table 5.

## 4 LIMITATIONS

In the proposed framework, we assume that the KG contains sufficient information to evaluate a given answer. In practice, this may not always be the case; therefore, we should consider the knowledge coverage of the KG. Effective routing to the most relevant KG for a given context could significantly improve the relevance

of the information retrieved. This indicates a need for further development in linking LLM queries to the most appropriate KGs. In order to support complex relationships, path prediction using knowledge graph embedding models [20, 22] or path ranking algorithms [12, 13] could be used. Additionally, the final quality score computed is based on hand-tuned parameters, which may lead to sub-optimal results. We expect to apply deep neural networks in the future to facilitate weight tuning. More sophisticated feature designs could be tried.

## 5 CONCLUSION AND FUTURE WORK

This work presents an automatic scoring system for answer quality that is reliable, interpretable, and faithful. Our scoring system is a meaningful attempt at combating the LLM hallucination problem. Our method opens up opportunities for subsequent applications like candidate answer ranking, results filtering, and more.

Future work involves extending testing to a wider array of LLM-generated answers and diverse standard QA datasets to better assess the scalability and robustness of our scoring system. We also plan to fine-tune the integration of LLMs with knowledge graphs, aiming to enhance the precision of evidence attribution and expand quantitative studies on benchmark results for a more comprehensive evaluation of the model's performance. By understanding the rationale behind LLM outputs at a more granular level, we expect to gain insights into the decision-making process of these models, potentially leading to more sophisticated applications.

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
