# OpenReview forum: "An Interpretable Answer Scoring Framework"
_ACM.org/SIGIR/2024/Workshop/Gen-IR — Gen-IR_SIGIR24_

### Official Review · Reviewer_Lk5E · 2024-05-24
**Reject - Tenuous connection to the workshop's topic, unclear presentation, unclear novelty**

**Rating:** -2
**Confidence:** 3

**Review:**

The paper presents a composite measure of answer quality as a weighted sum of multiple features:
1) Knowledge score: three subcomponents based on matching the answer's mentioned entities with a KB, as well as checking the entailment relation with an induced KB triplet subgraph 2) Structure score: presentation-level features used in some way (never explained in the paper) to create another component score 3) Content score based on coverage over the entities mentioned in the ground truth answer. Weights are chosen based on a "heuristic design". The experimental setup is basically a table of average scores on a subset of queries from the YahooQA dataset. No attempt was made to understand the correlation with human judgement.

I strongly recommend rejection.

Strengths:
- Timely topic.
- The paper is correctly formatted.
- The intent/structure relation is interesting.

Weaknesses:
- There are gaping holes in the presentation, so that it's hard to understand crucial details of the approach, such as the design of the Structure Score
- There is next to no motivation for crucial choices, such as the weights
- There is no real experimental setup to justify the need for the measure to begin with
- I fail to understand the connection between this paper and generative retrieval. The only thing that looks like retrieval is the entity linking steps require by the metric. This might not be a big issue if the workshop accepts submissions on grounding LM outputs to a KB.

---

### Official Review · Reviewer_bVf6 · 2024-05-26
**An Interpretable Answer Scoring Framework**

**Rating:** -1
**Confidence:** 3

**Review:**

Strengths：
1. The proposed interpretable scoring system evaluates answers based on knowledge, content, and structure, which is a novel approach to enhancing the quality and reliability of answers generated by LLMs.
2. The paper highlights the importance of explainability in information retrieval systems, providing a framework that makes the evaluation process transparent and understandable.
3. The detailed methodology, including the use of a knowledge graph for knowledge evaluation, heuristic features for structure, and entity alignment for content, is well-structured and thoroughly explained.

Weakness:
1. The paper could benefit from elaborating on the real-world applicability of the system with case studies or examples, and incorporating results from user studies or feedback to provide insights into the system's effectiveness and usability from an end-user perspective.
2. The proposed system’s implementation details are complex, and the paper does not provide sufficient guidance or tools for replicating the results, which may hinder adoption by other researchers or practitioners.
3. The paper does not address potential scalability issues that may arise when applying the scoring system to large-scale datasets or in high-demand environments. This is an important consideration for practical applications.

---

### Official Review · Reviewer_ohEx · 2024-05-27
**An interpretable answer scoring framework, but empirical (or theoretical) validation needed**

**Rating:** 1
**Confidence:** 2

**Review:**

Evaluating the output of LLMs is one prominent open problem in the area. The paper proposes an elaborate scoring scheme, accounting for the multi-faceted nature of answer evaluation. The proposed framework is rich and comprehensive, yet involves heuristic design and the question remains open how it can be validated. By its structured design I find it fair to say that the scoring framework is interpretable, but how do we know it is reliable and faithful? What exactly does the "experiment" described in section 3 prove?

---

### Decision · Program_Chairs · 2024-05-29

**Decision:**

Accept

**Comment:**

## ~Decision: Conditional Accept – Reproducibility~
## Decision: Accept [modified upon resubmission]

This paper proposes a heuristic answer scoring framework for evaluating LLM-generated answers along several dimensions (i.e., knowledge, structure, and content). Reviewers noted that the proposed framework is comprehensive and novel, but the heuristic design comes with trade-offs. Specifically, reviewers felt that it could be hard to reproduce or re-implement the work.

The chairs see merit in the proposed interpretable approach but share the reviewers' concerns regarding reproducibility, especially in the KG-AttrScore feature and NFQA taxonomy. Given this, we can provide the authors with two additional pages to describe the implementation in detail. We strongly encourage releasing code if at all possible.

Balancing the positive aspects of the paper with the reviewers' feedback, it is difficult to accept the paper directly as-is, but we would like to include it in the workshop. Therefore, the paper is accepted on the condition that, in the camera ready version, the implementation is clarified to the degree that it can be easily reproducible; either via a provided codebase (preferred, if possible) or thorough implementation details.

We propose the following process, to avoid giving the authors an additional deadline:
1. The paper is uploaded to the workshop (i.e., the non-archival camera ready is submitted) in around 10-15 days. The camera-ready deadline announcement will closely follow the decision announcement on June 30th.
2. We will check that the implementation has been sufficiently clarified. If our conclusion is negative, we unfortunately will not be able to include it in the workshop.

**In the meantime, the authors may provide an updated PDF to the organizers directly (gen-ir-sigir24@googlegroups.com) before the camera-ready deadline, if they would like early feedback.**